# Sea Fennel (*Crithmum maritimum* L.) as an Emerging Crop for the Manufacturing of Innovative Foods and Nutraceuticals

**DOI:** 10.3390/molecules28124741

**Published:** 2023-06-13

**Authors:** Maryem Kraouia, Ancuta Nartea, Antonietta Maoloni, Andrea Osimani, Cristiana Garofalo, Benedetta Fanesi, Lama Ismaiel, Lucia Aquilanti, Deborah Pacetti

**Affiliations:** Department of Agricultural, Food and Environmental Sciences (D3A), Università Politecnica delle Marche (UNIVPM), 60131 Ancona, Italy; m.kraouia@pm.univpm.it (M.K.); a.nartea@univpm.it (A.N.); a.maoloni@univpm.it (A.M.); a.osimani@univpm.it (A.O.); c.garofalo@univpm.it (C.G.); b.fanesi@pm.univpm.it (B.F.); l.ismaiel@pm.univpm.it (L.I.); d.pacetti@univpm.it (D.P.)

**Keywords:** halophyte, bioactive compounds, functional foods, functional food ingredients, nutraceuticals, edible film, essential oils

## Abstract

Sea fennel (*Crithmum maritimum* L.) is a perennial, strongly aromatic herb that has been used since ancient times in cuisine and folk medicine due to its renowned properties. Recently described as a “cash” crop, sea fennel is an ideal candidate for the promotion of halophyte agriculture in the Mediterranean basin due to its acknowledged adaptation to the Mediterranean climate, its resilience to risks/shocks related to climate changes, and its exploitability in food and non-food applications, which generates an alternative source of employment in rural areas. The present review provides insight into the nutritional and functional traits of this new crop as well as its exploitation in innovative food and nutraceutical applications. Various previous studies have fully demonstrated the high biological and nutritional potential of sea fennel, highlighting its high content of bioactive compounds, including polyphenols, carotenoids, ω-3 and ω-6 essential fatty acids, minerals, vitamins, and essential oils. Moreover, in previous studies, this aromatic halophyte showed good potential for application in the manufacturing of high-value foods, including both fermented and unfermented preserves, sauces, powders, and spices, herbal infusions and decoctions, and even edible films, as well as nutraceuticals. Further research efforts are needed to fully disclose the potential of this halophyte in view of its full exploitation by the food and nutraceutical industries.

## 1. Introduction

Sea fennel (*Crithmum maritimum* L.) is a perennial, strongly aromatic herb, commonly referred to as “Kritmo” in Greece or Rock Samphire, Samphire, and St. Peter’s herb in England, a name which derives from Saint Peter, the patron saint of fishermen, funcho do mar in Portugal, hinojo marino in Spain, criste marine in France, finocchio marino in Italy, motar in Croatia, kritamo in Greece, denizteresi in Türkiye, and shanar bahariya in Arabia. The generic term *Crithmum* derives from the Greek word “κρῑθή krithe”, meaning barley, due to the resemblance of the fruit to a grain of barley, while *maritimum* derives from maritime, indicating the growth of this plant near the sea. It is the only species in the genus *Crithmum* [1,2,3].

It is widespread in the coastal areas of Southern and Western European countries, along the coasts of the Mediterranean Sea, North America, Central and Western Asia. It grows spontaneously on cliffs, piers, gravel beaches, and rocky or sandy shores in the marine breeze [1,2,3] (Figure 1).

Sea fennel is a facultative halophyte, meaning that it is among the few species capable of surviving in saline environments (e.g., soils impregnated with seawater or regularly exposed to the splashing of sea waves).

However, the optimum growth of sea fennel occurs in salt-free or low-salinity environments. The plant has a robust rhizomatous root which can stretch up to five meters, and a branched stem, which is often woody at the base. The leaves are persistent, glabrous, with a triangular outline, bi or tri-pennate with fleshy and keeled lanceolate segments, and alternate with a long petiole which, when widening, forms a sheath, enveloping the base of the stem (Figure 2). Five-petaled and pale-yellow flowers are arranged in umbels, with 10–30 rays, subdivided in umbellets surrounded by bracts; the flowering period usually spans from June to September. Flowering plants produce achenes 3–6 mm in diameter, which are ovoid and glabrous with 10 ridges. The fruit begins to mature in November–December [1,2,3].

Since ancient times, sea fennel has been used in both cuisine and folk medicine as a tonic, carminative, vermifuge, diuretic, and anti-scurvy medicine. Though cited by Pliny the Elder in *Naturalis Historia* as early as the 1st-century a. C., the plant reached its maximum notoriety with the following quotation from *King Lear* by William Shakespeare “*Half-way down Hangs one that gathers samphire; dreadful trade*!”, Act IV, Scene VI, lines 14–15, where the immortal Bard refers to the dangerous practice of collecting the young sprouts of sea fennel near the cliffs overlooking the sea.

In most Mediterranean countries, including Spain, Slovenia, Croatia, Bosnia–Herzegovina, Greece, Cyprus, Türkiye, Tunisia, and Morocco, tender leaves and stems, mainly harvested in spring and summer up until early autumn, have been traditionally consumed in salads either raw or previously scalded in boiling water to soften [2]. They are also often preserved as pickles in olive oil, vinegar, or brine. In some regions across the Mediterranean basin, they are also consumed as an appetizer with various kinds of foods, with bread and olive oil, or even prepared with capers. The use of this halophyte for the preparation of condiments, such as cooked vegetables, seasoning olives with other aromatic herbs, and the homemade preparation of anchovies in brine, has also been reported. In addition, reports citing its exploitation to produce sauces, soups, fish condiments, and even an herb liqueur are also available [1,2].

The indiscriminate collection of sea fennel from the wild has led to the disappearance of this species from some European habitats. For this reason, in some regions, such as England and the Mt. Conero Natural Park (Marche Region, Central Italy), the plant is now protected, and its harvest from the wild is forbidden [4].

To date, *C. maritimum* L. is under used in Europe and the Mediterranean basin, though small- and medium-sized farms increasingly exploit this natural resource for crop production. In fact, recently described as a cash crop, sea fennel represents an ideal candidate for the promotion of halophyte agriculture in the Mediterranean basin due to its acknowledged adaptation to the Mediterranean climate, its resilience to risks/shocks related to climate changes (e.g., seawater intrusion, soil salinization, short-term water drought, etc.), and its exploitability in food and non-food applications, which generates an alternative source of employment in rural areas [5].

The present review focuses on the nutritional and functional traits of this highly aromatic herb as well as on its main bioactive molecules (e.g., phenolic compounds, essential oils, fatty acids, carotenoids, and vitamins) and its innovative food and nutraceutical applications. To this end, online databases (Scopus, https://www.scopus.com/, (accessed on 1 March 2023); Google Scholar, https://scholar.google.com/, (accessed on 1 March 2023); Science Direct, https://www.sciencedirect.com/, (accessed on 1 March 2023) were screened using “sea fennel”, “*Crithmum maritimum*”, “food”, “food ingredient”, and “nutraceutical” as keywords. A hierarchical approach based on the title, abstract, methods, and main results was adopted to select original research papers proposing innovative solutions at technology readiness levels (TRLs) of 2–5. Thus, we considered papers (i), which characterized the chemical profile and the bioactive compounds of sea fennel (TRL2–3), and (ii) developed laboratory-scale (TRL4) or industrial-scale food and food ingredient prototypes (TRL5). A database of 108 papers, with the year of publication ranging from 1999 to 2023, was established to extract information. Data were presented in Tables, and the values of the selected studies were expressed according to the most common unit of measurement to make comparisons among the results.

## 2. Health-Beneficial Compounds

Sea fennel represents a rich source of hydrophilic (polyphenols, vitamin C) and lipophilic (carotenoids, essential oils, fatty acids) bioactive compounds [6,7,8,9,10,11]. The latter are distributed differently along the plant (e.g., leaves, stems, flowers, and seeds), and they may find application in several sectors such as food, medicine, cosmetic, and various others [8]. Bioactive compounds are mainly secondary plant metabolites. In fact, the plant produces and accumulates these compounds as a defense or attractant under specific conditions, such as adverse climate conditions, nutrient deficiency, salinity, pest/predator attacks, etc. [7,12]. Moreover, their composition varies according to the genotype [13], habitat [14], and vegetative stage [7,15,16]. As an example, it was found that sandhill sea fennel individuals accumulate more chlorogenic acid than those growing on cliffs [17]. In another study, the application of methyl jasmonate (a growth regulator and modulator for plant defense responses) on sea fennel leaves to enhance toleration to high salinity (150 mM NaCl) was found to increase the fraction of unsaturated FAs, such as oleic and linoleic acids. Hence, the unsaturation of FAs was a consequence of the adaptation to high salt stress [14]. Exogenous stimuli were also adopted by Wang et al. [18] to promote the biosynthesis of monoterpene precursors, namely isopentenyl diphosphate and dimethylallyl diphosphate, with multiple beneficial biological activities. Given these premises, extraction methods and preservation techniques should be carefully evaluated to maximize the extraction of the target compound and to increase the shelf life of extracts, respectively.

### 2.1. Polyphenols

The polyphenolic profile of sea fennel (leaves, stems, flowers) is mainly characterized by the presence of hydroxycinnamic acids, predominantly chlorogenic acid (CGA), and flavonoids such as rutin and quercetin (Table 1) [19,20,21]. The main methods for their extraction and quantification involved hydro-alcoholic partitioning, followed by spectrophotometric methods or liquid chromatography coupled with a photodiode array and mass spectrometry detectors (Table 1). In general, CGA represents the most predominant polyphenolic compound, reaching 1.9–2.8% of dry weight (DW) [17]. However, the use of different solvent mixtures with different polarity indexes and extraction parameters (Table 1) affects the phenolic composition of the extracts, which in turn affects the antioxidant functional traits [16,22]. Among the CGA isomers, chlorogenic (3-O-caffeoylquinic acid), cryptochlorogenic (4-O-caffeoylquinic acid), and neochlorogenic (5-O-caffeoylquinic acid) acids were generally found as major metabolites in the leaf and stem tissue of sea fennel harvested in different regions [21,23,24,25,26]. However, in Greek sea fennel genotypes, the predominant quantified CGA isomer was 5-coumaroyl-quinic acid, followed by chlorogenic, 3,5-dicaffeoyl-quinic, 1-caffeoyl-quinic, 5-feruloy-quinic, cryptochlorogenic, and neochlorogenic acids [13].

Besides hydroxycinnamic acids, flavonoids were also identified in alcoholic extracts of sea fennel [13,21,27,28]. Among them, rutin and cirsiliol were the most abundant in ethanolic leaf extract [28]. Piatti et al. [20] identified nineteen phenolic compounds in the Italian sea fennel crop, with rutin, quercetin-3-O-galactoside, and luteolin-6,8-diglucoside being the most abundant flavonoids in the ethanolic leaf extract. Quercetin, rutin, and myricetin were also found in the flowers, stems, and leaves of Croatian wild sea fennel [21]. Differently, in hydro-methanolic extract, rutin and neochlorogenic acid were the major compounds [9], followed by vicenin-2 and other quercetin derivates [13]. In acetone extracts, rutin, catechin, and epigallocatechin were the predominant flavonoids [10]. Other compounds in minor quantities, such as gallic and vanillic acids, were present [15]. Sánchez–Faure et al. [6] identified twenty-five phenolic compounds in sea fennel, including minor ones such as apigenin. The total phenolic and flavonoid contents were concentrated as flowers > leaves > stems in wild Croatian sea fennel [21].

**Table 1 molecules-28-04741-t001:** The major classes of polyphenolic compounds in sea fennel extracts (in order of increasing polarity index) and food applications.

Source	Extraction	Chlorogenic Acid (mg g^−1^ DW)	TPC (mg g^−1^ DW)	TFC (mg g^−1^ DW)	Reference
Whole plant	Water (100%)	42.61 ^a^	N/A	N/A	[22]
	Ethanol (70%)	22.3 ^a^	N/A	N/A	[20]
	Ethanol (50%)	N/A	3.6	N/A	[6]
	Ethanol (40%)	58.48 ^a^	N/A	N/A	[22]
	Ethanol (40%)	N/A	23.44 ^a^	16.6 ^a^	[29]
	Ethanol (40%)	N/A	0.78 ^a^	0.49 ^a^	
	Methanol (80%)	6.36	47.1	17.3	[10]
	Methanol (80%)	N/A	2.59 ^b^	21.14 ^b^	[27]
	Methanol (70%)	2.58	2.55–10.84	2.25–15.08	[13]
	Methanol and hexane	N/A	8.11	56.20	[30]
	Acetone (80%)	N/A	4.1–7.9 ^a^	2.9–6.1 ^a^	[31]
Leaves	Ethanol (80%)	7.25 ^a^	31.7 ^a^	17.3 ^a^	[24]
	Ethanol (80%)	7.07–16.28	~16-22	N/A	[15]
	Methanol (80%)	N/A	26.3 ^a^	15.6 ^a^	[28]
	Methanol (70%)	1.4–4.8	~5.0–45.0	2.8–41	[9]
	Methanol (70%)	~3	N/A	N/A	[7]
	Methanol (50%)	~28	~32	N/A	[17]
	Methanol (100%)	N/A	0.9 ^b^	2.19 ^b^	[14]
	Acetone (80%)	N/A	7.16	4.77	[16]
	Acetone (80%)	N/A	8.27	3.45	
Seeds	Methanol (70%)	0.29	4.23–6.03	N/A	[32]
Essential oil	Distillation	N/A	7.5 ^d^	N/A	[33]
By product	Distillation	4.48–17.69 ^c^	70–150 ^c^	150–310 ^c^	[21]
	Distillation	13.67 ^a^	N/A	N/A	[21]
Infusion/decoction from leaves/stems/flowers	Water (100%)	8.24–8.67 ^e^	33.7–35.3 ^e^	54.4–57.2 ^e^	[23]
Fermented leaves	Methanol (80%)	N/A	0.77 ^b^	3.9 ^b^	[27]
	Methanol (80%)	N/A	0.4	N/A	[34]

The reported data were adapted to ensure the uniformity of units between the reviewed studies. ^a^ mg g^−1^ DW extract; ^b^ mg g^−1^ FW; ^c^ mg g^−1^ extract; ^d^ mg g^−1^ essential oil; ^e^ mg 200 mL^−1^ cup. Chlorogenic acid (3-caffeoylquinic acid); TPC: total polyphenol content; TFC: total flavonoid content; DW: dry weight; FW: fresh weight; N/A: not available.

### 2.2. Essential Oils (EOs)

Sea fennel essential oils (EOs) consist of volatile organic compounds that are secondary metabolites produced by the plant to protect itself from adverse conditions as well as to attract pollinators [35]. Hydro-distillation is the most common and straightforward practice used to extract sea fennel EOs. Beyond the extraction method, the organs of the plant used may influence the yield and profile of EOs [21]. Among the papers we have reviewed, the EO yield was always inferior to 1%, with few exceptions registered in flowers [21], flowering aerial parts [36,37], and seeds [38] (Table 2).

The major classes of compounds in sea fennel EOs are monoterpene hydrocarbons, oxygenated monoterpenes, and phenylpropanoids (Table 2). Within the first group, sabinene and γ-terpinene were the most characteristic compounds in EOs from different genotypes, the former compound varying from 0.2 to 51.5%, discovered by analyzing the EOs extracted from the leaves, while the latter compound varying from 2.8 to 50.5%, detected by analyzing the EOs extracted from the whole plant at the flowering stage. Except for a few studies, limonene, β-ocimene, p-cimene, α-pinene, and α/β-phellandrene were infrequently identified. For example, limonene > sabinene > γ-terpinene > α-pinene > β-ocimene is the order which characterized the EOs hydro-distilled from the aerial part of Croatian wild sea fennel at the flowering stage [26]. Oxygenated monoterpenes mainly include terpinene-4-ol, carvacrol methyl ether, and thymol methyl ether. The first two were absent in the methanolic extracts analyzed by Castillo et al. [7], whereas thymol methyl ether was the major terpenoid found at high concentrations (up to 89%). Dillapiole was the most representative phenylpropanoid. Depending on the sea fennel genotype, the amount of dillapiole varied greatly from 0 to 64.2% (detected in the EOs obtained from sea fennel leaves). Accordingly, each genotype may respond differently to stresses such as salinity, nutrient availability, etc. [7,39]. Notably, the EOs obtained from the same plant material extracted by using two different methods (e.g., hydro-distillation and supercritical fluid extraction) did not show any significant variations in terms of volatile profile [40]. On the contrary, microwave-assisted hydro-distillation gathered EOs from sea fennel with a higher yield of limonene and sabinene than the conventional method of hydro-distillation [26].

**Table 2 molecules-28-04741-t002:** *Crithmum maritimum* L. essential oil yield and composition based on country of origin, extraction method, and plant material used.

Source	Extraction	Oil Yield (%)	Monoterpene Hydrocarbons (%)							Oxygenated Monoterpenes (%)			Phenylpropanoids (%)	Reference
			Sabinene	γ-Terpinene	Limonene	β-Ocimene	p-Cimene	α-Pinene	α/β-Phellandrene	Terpenen-4-ol	Carvacrol Methyl Ether	Thymol Methyl Ether	Dillapiole	
Whole plant	SD	N/A	49.5	31.4	2.7	N/A	0.6	9.6	N/A	1.5	N/A	N/A	N/A	[33]
	SD	0.2	5	50	8.9	0.31	8.9	2.6	0.2	0.6	0.3	18.2	2.5	[20]
	HD	0.2	0.5	19.3	N/A	N/A	5.8	1	6.6	2.3	N/A	20.6	40.3	[31]
	HD	0.3	2.2	30.6	N/A	N/A	9.9	1.6	N/A	0.4	N/A	40.4	14.3	
	HD	0.1	22.3	28.4	12.1	8.9	2.5	7.1	10.7	1.2	2.6	0.2	N/A	[27]
	HD + ME	N/A	N/A	33	N/A	N/A	8.7	N/A	N/A	N/A	N/A	22	17.5	[41]
Whole plant (flowering period)	HD + ME	0.8	N/A	32.9	N/A	N/A	N/A	N/A	N/A	N/A	21.9	N/A	17.5	[42]
	HD	N/A	25.2	7.1	51.4	4.0	1.8	4.3	N/A	1.8	N/A	N/A	N/A	[26]
	MHD	N/A	27.8	6.5	53.1	3.0	2.5	2.3	N/A	1.3	N/A	N/A	N/A	[26]
	HD	2.3	35.6	18.7	N/A	1.6	N/A	0.9	22.5	3.1	N/A	10.9	N/A	[36]
	HD	0.6	12.4	19.9	38.4	5	2.6	1.8	N/A	3.1	4.2	0.1	8.1	[43]
	HD	0.1	0.7	50.5	N/A	N/A	12.6	0.3	N/A	N/A	0.1	33.7	N/A	[10]
	HD	N/A	26.5	2.8	58.4	N/A	N/A	1.5	N/A	5.6	N/A	N/A	N/A	[44]
	HD	0.4	N/A	>10	N/A	>10	N/A	N/A	>10	N/A	N/A	>10	N/A	[45]
	HD	N/A	32	33.6	0.2	2.8	3.9	1.2	0.6	3.4	N/A	15.7	0.1	[46]
Leaves (fermented)	HD	0.1	1.3	42.1	7.5	9.6	5.7	3.5	12	2.8	8.6	0.4	N/A	[27]
Leaves	UAE	N/A	8.1–25.8	N/A	0.1	N/A	0.7	0.2	0.2			67.7–89.1		[7]
	HD	0.6	13.4	12.0	57.5	N/A	N/A	0.1	N/A	6.9	N/A	0.3	N/A	[37]
	HD	0.6	51.5	3.5	36.3	N/A	0.1	0.1	N/A	5.4	N/A	N/A	N/A	[21]
	HD	0.5	0.4	22.5	N/A	N/A	4.8	N/A	N/A	0.2	N/A	27.8	41.4	[38]
	HD	N/A	0.4–16.2	27.1–41.5	N/A	1.8–2.1	5.2–7.5	0.8–5.9	0.1–13.3	0.3–3.5	N/A	2.3–14.9	0.2–41	[40]
	SFE	N/A	0.2–9.3	14.4–40.1	N/A	1.3–1.9	4.1–11.8	0.3–4.9	0.1–6.1	0.2–1.8	N/A	2.3–22.9	0.2–64.2	
Flowers	HD	2.4	12.0	13.8	62.2	N/A	N/A	4.9	N/A	2.0	N/A	0.2	N/A	[37]
	HD	1.4	44.9	2.8	43.6	N/A	N/A	1.8	N/A	3.5	N/A	N/A	N/A	[21]
	HD	0.3	0.7	43.3	N/A	N/A	14.7	0.9	0.1	0.3	N/A	34.3	N/A	[38]
Stems	HD	0.2	8.1	4.6	74.2	N/A	N/A	0.1	N/A	5.9	N/A	0.4	NA	[37]
	HD	0.3	0.5	32.8	N/A	N/A	5.9	1.1	0.2	0.2	N/A	26.8	31	[38]
	HD	0.6	42.6	5.3	36.5	N/A	0.3	0.4	N/A	10.4	N/A	N/A	N/A	[21]
	HD	0.6	43	10	N/A	N/A	13	10	13	N/A	N/A	N/A	N/A	[47]
Seeds	HD	0.6	43.3	10.6	N/A	N/A	13.3	10.6	13.2	1.2	0.1	2.1	N/A	[47]
	HD	3.6	1.0	39.7	N/A	N/A	26.1	N/A	N/A	1.1	0.1	20.1	7.9	[38]

SD: steam distillation; HD: hydro-distillation; ME: microemulsion; MHD: microwave-assisted hydro-distillation; SFE: supercritical fluid extraction with CO_2_; UAE: ultrasound-assisted extraction (with methanol as organic solvent); N/A: not available.

### 2.3. Fatty Acids

Lipids may be considered primary or secondary metabolites because they can regulate several biological functions within plant cells. Temperature and other environmental factors might influence their biosynthesis [48,49]. Sea fennel exhibited a promising fatty acids profile from both a nutritional point of view, for the presence of unsaturated fatty acids (e.g., oleic, linoleic, and linolenic acid), and an industrial perspective, bearing in mind the relative abundance of petroselinic acid (an antiaging agent) [32,50,51].

The fatty acid profile of leaves [7,9,14,39,52], sprouts [34], and stems [6] are dominated by polyunsaturated fatty acids (PUFAs), which ranged from 42.0 to 86.0% of total fatty acids, whereas saturated (SFAs) and monounsaturated (MUFAs) fatty acids varied from 11.8 to 35.3% and from 2.3 to 32.0%, respectively (Table 3). Among PUFAs, the main fatty acids are linoleic and linolenic acids. Oleic and palmitic acids are the most abundant among MUFAs and SFAs. Sánchez–Faure et al. [6] analyzed the composition of lipid fractions in lyophilized stems and indicated that the sea fennel fatty acid profile was PUFAs > SFAs > MUFAs, while in seeds, the distribution of the fractions was MUFAs > PUFAs > SFAs [32,50].

For the ω6/ω3 ratio (Table 3), the highest value (1.6) was recorded in fermented blanched sea fennel sprouts destined for the manufacturing of innovative preserves [34]. Sánchez–Faure et al. [6] found a ω6/ω3 ratio equal to 1.12 by analyzing freeze-dried sea fennel leaves; neatly lower ω6/ω3 ratios were recorded by the same authors in other halophytes, being common ice plant (Carpobrotus chilensis) and seaside arrowgrass (Triglochin maritima), with values equal to 0.41 and 0.42, respectively, both in the range of nutritional recommendations for excellent health benefits [6,53,54]. Very recently, Maoloni et al. [55] first detected two hydroxylated fatty acids, namely trihydroxy-octadecadienoic acid and trihydroxy-octadecenoic acid, in Italian sea fennel cultivated in central Italy. Both compounds are characterized by antineuroinflammatory, cytotoxic, and antimicrobial activities [56,57].

### 2.4. Vitamins

Halophytes are known to be a source of both hydrophilic (e.g., C, B group) and lipophilic vitamins (e.g., A, E) [9,58,59,60,61]. However, except for vitamin C, to the best of the authors’ knowledge, no other vitamins have been investigated in sea fennel yet.

A high content of vitamin C, ranging from 39.0 to 76.6 mg 100 g^−1^ wet weight (WW), was reported for fresh sea fennel leaves [2]. A vitamin C content attesting at 51 mg 100 g^−1^ WW was found by Maoloni et al. [34] in fresh leaves collected from Italian sea fennel crop, whereas in laboratory-scale prototypes of preserves made with blanched sprouts subjected to fermentation, vitamin C was under the limit of detection. This finding was ascribed by the authors to the peculiar features of this water-soluble and heat-labile vitamin. In another study, a sea fennel aqueous extract containing 1.46 mg g^−1^ dry weight (DW) of vitamin C was encapsulated in liposomes to protect this vitamin from degradation [22]. Regarding the vitamin B group, it has been reported that Mesembryanthemum nodiflorum, Suaeda maritima, and Sarcocornia fruticosa were rich in vitamin B6, with higher values than watercress and asparagus vegetables (2.63 and 1.34 mg 100 g^−1^ DW, respectively) [61]. Vitamin B6 and B1 were quantified in Salicornia ramosissima cultivated at six salinities, at ranges from 0.3 to 2.6 and 4.3 to 30.4 mg 100 g^−1^ WW, respectively [60].

To date, sea fennel carotenoid matter, containing compounds with provitamin A activity [62], has only been investigated using spectrophotometric methods [10,30]. The latter allows the total amount of carotenoids to be quantified without being aware of the provitamin A activity. The total content of carotenoids ranging from 33.0 to 56.0 mg kg^−1^ WW of fresh leaves, with a mean value of 44.6 mg kg^−1^ WW, was reported by Tardio et al. [2]. Nabet et al. [10] reported 62.2 mg kg^−1^ DW, which is about 7.5-fold lower than that found by Sousa et al. [30]. Moreover, extraction with methanol allowed a total carotenoid content of ~2.43 mg kg^−1^ DW to be detected in sea fennel leaves [14]. On the other hand, Guil-Guerrero and Rodríguez-García [63] reported a much higher level of carotenoids (338 mg kg^−1^ DW of tender leaves) using a chloroform–methanol extraction procedure [63].

### 2.5. Minerals

According to various reports [9,64], sea fennel is a good source of minerals, especially Ca, Mg, and K. Among these elements, Ca and Mg are of key importance for human nutrition, especially for cellular metabolism, as well as bone structure and development. In detail, a mean leaf content of Ca ranging from 2.2 to 3.7 g 100 g^−1^ DW was reported by Martins–Noguerol et al. [9]. For Mg and K, mean contents ranging from 0.4 to 0.6 g 100 g^−1^ DW and from 1.8 to 6.0 g 100 g^−1^ DW were reported by the same authors. For Ca, sea fennel was also reported to be a better source than broccoli, the latter representing one of the vegetable sources with the highest content of this mineral [65].

Regarding microelements, data collected by Martins–Noguerol et al. [9] indicated that sea fennel is a source of Fe, Mn, Zn, Cu, and Cr, with mean contents ranging from 191.7 to 58.4 mg kg^−1^ DW, 80.3 to 37.2 mg kg^−1^ DW, 23.5 to 41.3 mg kg^−1^ DW, 2.8 to 7.3 mg kg^−1^ DW, and 0.5 to 2.7 mg kg^−1^ DW, respectively, depending on both the type of habitat and the edaphic conditions.

## 3. Compounds with Adverse Effects

Despite the high nutritional composition of sea fennel, it is necessary to consider the relative content of some antinutrients occurring in this halophyte, such as tannins [21] and soluble fiber [6,34]. Hydro-distillation by-products resulting from EO extraction were found to contain the highest levels of total condensed tannins (TCTs) (200 mg g^−1^ extract), whereas <1 mg g^−1^ DW was found in extracts prepared from the aerial parts of sea fennel [10,23,24,31]. These antinutrients may cause mineral deficiencies and micronutrient malnutrition. Hence, they can precipitate proteins and chelate iron and zinc, thus reducing the absorption of cations [66]. To date, various processing methods have been developed and applied, either separately or in combination (e.g., fermentation, soaking, autoclaving, etc.), to reduce this negative effect [66]. As an example, in a recent study aimed at searching for new sources of innovative products for the food industry within halophytes, a low content of condensed tannins was found in aqueous sea fennel extracts (infusion and decoction) [23]; analogously, a low content of these compounds was found by Jallali et al. [31] in sea fennel acetonic extracts.

Sea fennel is also rich in dietary fiber, attesting between ~3.0 and ~6.2 g 100 g^−1^ WW [34]. As a rule, a high intake of dietary fiber negatively affects the absorption of carbohydrates, thus leading to lower blood glucose and insulin; however, an excessive intake of dietary fiber can also reduce the assimilation of minerals and favor gastrointestinal diseases due to fermentation and gas formation.

Regarding minerals with potential adverse effects, Na is the most abundant element in sea fennel leaves, with a mean content attesting to 14.7 g kg^−^^1^ WW according to Sánchez–Faure et al. [6], and ranging from 1.2 to 5.0 g 100 g^−^^1^ DW depending on the specific growth conditions according to Martins–Noguerol et al. [9]. In halophytes, and hence sea fennel, Na is usually accumulated in vegetable tissues as a direct consequence of soil salinity. Na is an essential nutrient for human beings; however, an excessive intake of this element is associated with an increase in blood pressure and, hence, with an augmented risk for cardiovascular diseases [67].

## 4. Functional Traits

In the last decade, there has been growing interest in both the food and pharmaceutical industries towards the natural compounds contained in plant extracts able to promote health and reduce the risk of disease. Like other halophytes, *C. maritimum* is a well-known source of compounds, such as polyphenols, with biological activity, which justifies its recognition as a potential functional food [61], the latter being a food offering health benefits beyond its nutritional value.

### 4.1. Antioxidant Activity

Different tests, such as DPPH (2,2-Diphenyl-1-picrylhydrazyl), ABTS (2,2′-azino-bis-3-ethylbenzothiazoline-6-sulfonic acid), ORAC (Oxygen Radical Absorbance Capacity), and FRAP (Ferric Reducing Antioxidant Power), have been used to evaluate the antioxidant properties of sea fennel extracts. High radical scavenging activities on DPPH and ABTS for the methanol/water extracts of sea fennel leaves were correlated to a high presence of phenolic compounds and mainly to chlorogenic acids [17]. Accordingly, Souid et al. [24] described the strong antioxidant activity of hydro-ethanolic (20:80) extract from sea fennel leaves for DPPH (0.22 IC_50_ mg mL^−1^), ABTS (2.07 mg Trolox equivalents g^−1^), ORAC (15.84 μmol Trolox equivalents g^−1^ DW) and FRAP (1.82 EC_50_ mg mL^−1^). Expanding the study to the ex vivo level, the same extract assured its trait as an antioxidant agent when it was exploited to treat human erythrocytes exposed to a strong oxidative insult (AAPH = 50 mM) during a cellular antioxidant activity assay and hemolysis test [24].

Unlike sea fennel hydro-alcoholic extracts, EOs from this halophyte herb resulted in weak radical scavenging activity, according to DPPH tests [44]. Jallali et al. [31] reported a low radical scavenging activity on DPPH (IC_50_: 0.059 ± 0.004; 0.146 ± 0.004 mg mL^−1^). This difference can be attributed to the different chemical compositions; the hydro-alcoholic extracts were mainly formed by chlorogenic acids, while the essential oils mainly contained terpenes [33]. However, sea fennel EOs showed a high capacity to scavenge the radical of linoleic acid peroxide according to the β-carotene bleaching method (IC_50_: 1.12 ± 0.041; 1.16 ± 0.063 mg mL^−1^), as well as high antioxidant activity, according to a TBARS (Thiobarbituric acid reactive substance) assay, for an EO concentration equal to 1 g L^−1^ [44].

### 4.2. Antimicrobial Activity

Meot–Duros et al. [68] reported the strong antimicrobial activity of a chloroform (apolar) extract from sea fennel leaves mostly against *Micrococcus luteus* and *Salmonella Arizonae* with a minimum inhibitory concentration (MIC) corresponding to 1 µg mL^−1^, but also against *Bacillus cereus* and *Candida albicans* with a MIC of 10 µg mL^−1^ and against *Erwinia carotovora*, *Pseudomonas marginalis*, *Pseudomonas fluorescens*, and *Escherichia coli* with a MIC of 100 µg mL^−1^ [68]. Lately, Meot–Duros et al. [69] isolated falcarindiol, a polyene with known antimicrobial properties, from chloroform extracts with antimicrobial activity (MIC corresponding to 50 µg mL^−1^) against *Micrococcus luteus* and *B. cereus*. In a further study, the EOs extracted from sea fennel displayed strong antimicrobial activity against *Staphylococcus aureus* and *B. cereus*, less activity against *Pseudomonas aeruginosa*, and no activity against *E. coli* [31]. Pedreiro et al. [70] also reported the high inhibition of sea fennel EOs against *Lactiplantibacillus plantarum*, *B. cereus*, *S. aureus*, and *E. coli*. Additionally, Campana et al. [42] described the antimicrobial activity of sea fennel EOs against *E. coli*, *Listeria monocytogenes*, *S. aureus*, *P. fluorescens*, and *C. albicans*. Moreover, the latter authors reported stronger antimicrobial activity of even microemulsions prepared with sea fennel EOs (5% *v v*^−1^) in a mixture of ethanol, glycerol, polysorbate, and water. Regarding methanolic (polar) extracts, they were found to be active against *Salmonella arizonae*, *P. aeruginosa*, and *P. fluorescens* (MIC of 1 µg mL^−1^), but also against *Pseudomonas marginalis* (MIC of 100 µg mL^−1^) [68]. Beyond sea fennel EOs, Jallali et al. [31] also investigated the antimicrobial activity of the acetonic extracts of sea fennel aerial parts. The latter showed neat inhibitory activity against *E. coli* and *S. aureus*, but no activity against *P. aeruginosa* and *B. cereus*.

### 4.3. Anti-Inflammatory, Anticarcinogenic, and Other Functional Activities

It is well known that phenolic compounds possess strong antioxidant, anti-inflammatory, and antiproliferative activities against cancer cells [71,72]. Chlorogenic acids, the main antioxidant compounds in sea fennel, are reported to have different autoimmune or inflammatory properties to face the hypoglycemic and hypolipidemic effects caused by diseases such as rheumatoid arthritis and diabetes mellitus [11]. According to Alemàn et al. [73], sea fennel aqueous and ethanolic extracts, especially rich in chlorogenic acids, showed anti-inflammatory activity, with both extracts inducing the secretion of IL-10 cytokines in macrophages. Moreover, sea fennel extracts encapsulated into soy phosphatidylcholine liposome downregulated the secretion of TNF α cytokine, and, thus, inflammation in macrophages previously stimulated with lipopolysaccharide. Notably, the exposure of vasoconstriction-induced rat aortic rings to sea fennel flower ethanolic extracts caused vasodilation, attributed at least partially to chlorogenic acid [37]. Moreover, the administration of a suspension of *C. maritimum* leaves in rats showed a hepatoprotective effect against carbon-tetrachloride-induced liver damage since the hepatic activities of some P450 enzymes were restored while oxidative stress was reduced [28]. Sea fennel extract may play a role in stress-related diseases as it was able to protect in vitro cells against heat-shock-induced stress [74]. Four flavonoids, isolated for the first time in sea fennel from Tayuan, demonstrated an antiproliferative effect against A375 melanoma cells responsible for skin cancer [75]. Considering that chlorogenic acids deriving from coffee ethanolic extracts were found to inhibit carbohydrate-hydrolyzing enzymes, such as α-amylase and α-glucosidase [11,76], the hypoglycemic effect could also be evaluated in sea fennel polar extracts. To date, no literature data exist concerning this effect.

Regarding sea fennel EOs, particularly rich in monoterpenes, anti-inflammatory potential has been demonstrated in in vitro models. At concentrations as low as 3.125 μg mL^−1^, the EO from sea fennel still decreased nitric oxide production by 37%, without showing toxicity, in lipopolysaccharide-stimulated macrophages [46]. EOs from flowers or leaves or the stem inhibited acetylcholinesterase and butyrylcholinesterase enzymes responsible for the neurodegenerative Alzheimer’s disease [37]. In another study, Kamte et al. [43] tested a sea fennel EO as a model for the identification of trypanocidal compounds to be used in alternative/integrative therapies for the treatment of Human African Trypanosomiasis (HAT), also called “African sleeping sickness”. These authors demonstrated that the whole oil and each monoterpene contained in such an EO had high selectivity against *Trypanosoma brucei*, the causative agent of HAT, following the order α-pinene > β-ocimene > p-cymene > limonene > sabinene > α-phellandrene > γ-terpinene > whole oil. Gnocchi et al. [77] successfully proved the role of sea fennel ethyl acetate extract in inhibiting the growth of hepatocellular carcinoma cells, indicating the antitumor activity of this halophyte plant. However, the EOs of *C. maritimum* did not show a significant antiproliferative effect against breast and colorectal cancer cells [45]. Finally, sea fennel hydro-alcoholic extract was active in preventing mutation induced by the oxidative damage caused by H_2_O_2_ in growth experiments [24].

## 5. Exploitation of Sea Fennel for the Manufacturing of Innovative Foods and Food Ingredients

Though sea fennel has been used for centuries as a food ingredient and even as medicine for phototherapeutic treatment, its exploitation for the manufacturing of innovative foods, food ingredients, and nutraceuticals has not been extensively examined [78,79]. In numerous Mediterranean countries, such as Italy, Greece, and France, sea fennel is consumed fresh in salads or as unfermented preserves in olive oil, brine, or in an aqueous solution of wine vinegar. Most commercially available preserves and sauces (including pesto-like ones) are manufactured on an industrial scale and often pasteurized. Such a thermal process, which guarantees the safety of the final product, dramatically affects the content of beneficial non-thermotolerant nutrients, such as vitamin C. Accordingly, in recent years, research efforts aimed at valorizing this new food crop through the application of mild processing technologies (e.g., fermentation, mild in-container pasteurization, drying, etc.) more respectful of the high biological and nutritional potential of sea fennel have been utilized to encourage the exploitation of this emerging halophyte plant. In the following sub-sections, the available studies focused on the exploitation of sea fennel to produce innovative foods and food ingredients are described in detail.

### 5.1. Fermented Preserves

The use of a thoroughly adapted starter is crucial to driving the fermentation of a hostile substrate such as sea fennel, characterized by its high antimicrobial activity against a wide range of microorganisms, including lactic acid bacteria [70,80,81].

A pioneering study by Özcan [78] explored the exploitation of sea fennel to produce fermented preserves in brine with and without the inoculation of yogurt as a starter culture under different conditions. As a main result, higher lactic acid bacteria activity, and hence a higher acidity increase, was seen in brine containing 8% salt, 1% sugar, and 1% yogurt compared to uninoculated brine containing purely 8% salt.

In 2019, Özcan et al. [27] carried out a study aimed at determining the influence of fermentation on sea fennel antioxidant activity, total phenolic, total flavonoid, and essential oil content. To this end, sea fennel aerial parts were fermented at 24 °C for 45 days in 10% brine containing citric acid, NaCl, ascorbic acid, potassium sorbate, and sodium benzoate to provide equilibrium values of 0.50 g 100 mL^−1^ titratable acidity (expressed as lactic acid), 4.7 g 100 mL^−1^ NaCl, 0.4 g L^−1^ ascorbic acid, 0.5 g L^−1^ sorbic acid, and 0.5 g L^−1^ benzoic acid, respectively. Based on the collected results, fresh sea fennel was recognized by the authors as a significant source of antioxidants and phenolic and flavonoid compounds, whereas fermentation led to the degradation of the plant’s phenolic compounds. Moreover, an increase in γ-terpinene, carvacrol methyl ether, 1,2-dihydroxybenzene, trans-cinnamic acid, and isorhamnetin was seen during fermentation.

Two years later, Maoloni et al. [34] formulated a multiple-strain starter for the pilot-scale manufacturing of innovative fermented sea-fennel-based preserves in brine containing 7% NaCl and 1% fructose (Figure 3). To this end, 27 lactic acid bacteria isolates, previously isolated from vegetable sources and ascribed to the species Lactiplantibacillus plantarum, Leuconostoc pseudomesenteroides, Pediococcus pentosaceus, and Weissella confuse [81], were preliminary assayed for their key protechnological (acidification rate and extent) and sensory (production of pleasant scents) traits in an ad hoc formulated sea-fennel-based medium [34,82]. Hence, five multiple-strain starters were formulated and assayed on a laboratory scale to produce prototypes of pickled sea fennel by fermentation at 30 °C until a fixed pH value of approximately 3.8 was reached. The best multiple-strain starter in terms of a fast drop in pH and the production of pleasant odorous notes was selected for the pilot-scale manufacturing of pickled sea fennel. To this end, blanched sea fennel sprouts were fermented in steel casks at room temperature (∼18–20 °C) until a fixed pH value of 3.80 was reached (approximately 9 weeks after inoculation). The analysis of the pilot-scale prototypes confirmed the validity of the physio-chemical data collected on the laboratory-scale prototypes, except for slower acidification due to the lower fermentation temperature. More specifically, a high content of polyunsaturated fatty acids, those being linoleic and α-linolenic acids, and of dietary fiber was highlighted. For the latter trait, according to Regulation (EC) No 1924/2006, the claim “source of fiber” was suggested by the authors for pickled sea fennel. Regarding the sensory analysis, a kerosene-like note and herbal scents, likely related to the volatile compounds p-cymene and α-pinene, were perceived in the pilot-scale prototypes by a trained panel of assessors, together with a salty flavor and a crunchy consistency. In agreement with what was reported by Özcan et al. [27], fermentation led to a decrease in total polyphenols and antioxidant activity. A significant decrease in the content of vitamin C (below the detection limit) was also seen during fermentation, the latter evidence being ascribed by the authors to the progressive degradation of such a vitamin due to residual oxygen occurring in the steel casks during fermentation.

One year later, the same multiple-strain starter was exploited by Maoloni et al. [7] for the co-fermentation of sea fennel aerial parts and green olives (*Olea europaea* L. cv. Ascolana tenera) [83]. To this end, the effects of two recipes [A: 10% sea fennel and 90% green olives; B: 60% sea fennel and 40% green olives] and two conventional methods for the production of table olives (Spanish- or Greek-style methods) were assayed in started and unstarted (control) laboratory-scale prototypes (Figure 4). During fermentation, all the prototypes showed a progressive reduction in pH. However, the recipe causing a drop in pH was hypothesized by the authors, with more intense acidification being recorded in the prototypes containing the highest percentage of sea fennel. In contrast, a faster reduction in the viable counts of Enterobacteriaceae was seen in the started prototypes with respect to unstarted ones. Moreover, an evolution of the microbiota was seen during fermentation by the metataxonomic analysis, with Lactiplantibacillus plantarum dominating all the prototypes in the late stage of fermentation, irrespective of the recipe, processing method, and starter addition. Regarding sensory analysis, the prototypes manufactured according to the Greek-style method were those most appreciated by a panel of trained assessors, feasibly due to the higher perceived crunchiness and prolonged fermentation, which notoriously leads to a high accumulation of odor- and flavor-related metabolites [69,70].

### 5.2. Artificially Acidified Preserves

Very recently, sea fennel was recognized as a promising ingredient for the formulation of innovative non-dairy probiotic foods. In this regard, Maoloni et al. [84] evaluated the survival and stability of both a commercially available formulation of human probiotics, SYNBIO^®®^, and *Lactiplantibacillus plantarum* IMC 509 during prolonged storage of a sea fennel preserve under refrigerated conditions. Briefly, blanched sea fennel sprouts were soaked in brine (containing 7% NaCl and 1% fructose and artificially acidified with 0.5% food-grade lactic acid), pasteurized at 95 °C for 5 min in boiling water, cooled in iced water, and stored at room temperature (18 ± 2 °C) for 4 weeks to allow the pH to equilibrate. Once the equilibrium pH of 3.85 ± 0.07 was reached, the probiotic strain *Lactiplantibacillus plantarum* (syn. *Lactobacillus plantarum*) IMC 509 and the formulation SYNBIO^®®^, a combination (1:1) of *Lacticaseibacillus rhamnosus* (syn. *Lactobacillus rhamnosus*) IMC 501^®®^ and *Lacticaseibacillus paracasei* (syn. *Lactobacillus rhamnosus*) IMC 502^®®^, were individually inoculated into the brine, and their viability was evaluated during refrigerated storage (4 °C) of 44 days. Viable counts of mesophilic lactic acid bacteria were high above the threshold recommended to exert beneficial effects on human health [85] for both the brine and drained sea fennel sprouts; this finding was explained by the authors due to the capability of the probiotic strains assayed to adapt by adhering to sea fennel tissue and hence adapting to this peculiar environment [84]. Thus, sea fennel was indicated by the authors to be a good substrate for formulating new non-dairy probiotic products, including the probiotic strain *L. plantarum* IMC 509 and the formulation SYNBIO^®®^.

### 5.3. Shelf-Stable Green Sauces

Sea fennel was also proposed by Maoloni et al. [86] as an ideal ingredient for the manufacture of industrial-scale prototypes of two new shelf-stable green sauces (Figure 5). In such a study, accelerated shelf life and microbial challenge tests were performed to assess (i) the microbiological shelf-stability of the two sauces stabilized by heat treatments commonly applied on an industrial scale to inactivate vegetative cells of spoilage microorganisms and pathogens in vegetable preserves (F_85_^7^ = 2 min or F_95_^7^ = 5 min), and (ii) the inhibition of Staphylococcus aureus and *Bacillus cereus* in the sauce with a pH = ∼ 4.6 and aw < 0.92, subjected to mild pasteurization (F_75_^7^ = 1 or 2 min). Overall, the collected results, through the accelerated growth tests carried out on the two newly developed green sauces subjected to industrial-like heat treatments equivalent to 85 °C for 2 min or, alternatively, 95 °C for 5 min (estimated based on Z = 7 °C), demonstrated the shelf-stability of both sauces for 1 month of storage under conditions of thermal abuse. Moreover, microbial challenge tests carried out on the sole sauce with a pH = ∼4.6 and aw < 0.92 with *S. aureus* revealed that both the mild heat treatments assayed (F_75_^7^ = 1 or 2 min) were able to kill the vegetative cells of this foodborne pathogen. In addition, inhibition of the growth of *B. cereus* was seen during storage at 37 °C of the same sauce challenged with this foodborne pathogen.

### 5.4. Powders

The preparation of powders from vegetable sources involves the removal of water from vegetable tissues by air- or freeze-drying, or the milling process, which in turn affects the particle size of the powder, thus the water absorption capacity and the solubility of phytochemicals [87]. In fact, below a particle size of 0.5 mm, the antioxidant activity of powders increases, as has been demonstrated for wheat bran by-products [88]. Powders obtained from the aerial parts of sea fennel (Figure 6) can be used as a new spice in culinary preparations and seasoning blends or as flavoring and colorant in pasta, fish-based foods, meatballs, rice, vegetable dishes, and sauces [79,89]. In fresh sea fennel leaves, color is mainly provided by carotenoids and chlorophylls, while aroma is mainly provided by EOs. Carotenoids can undergo degradation, isomerization, and oxidation during thermal treatments [90], while air-drying can reduce the intensity of the green color due to the consequent large decrease in the chlorophyll content [91]. To select the best treatment to produce sea fennel powders with high-quality sensory traits, both air- and freeze-drying were explored by Renna and Gonella [79]. As a result, the freeze-dried powder was characterized by a similar color to fresh sea fennel leaves, while air-dried powder showed a darker color than the freeze-dried one. In agreement with these preliminary results, Giungato et al. [92] found that the color of sea fennel was darkening as the air-drying temperature increased from 40 to 60 °C. In a further investigation, sea fennel treatment using microwave-assisted or freeze-drying methods reduced changes in color during the removal of water more than conventional air-drying and microwave-assisted air-drying [89]. Differences regarding odor and taste were also found in sea fennel spices produced by the heat treatment of fresh leaves in a forced-draft oven at 65 °C or using a freeze-drying method. In more detail, regarding taste, the freeze-dried powder was characterized by an initial note of herbs and freshness that, later on, turned into a celery flavor with a hint of salt, whereas the hot-air-dried powder showed less herbal flavor with a hint of salt [79].

### 5.5. Infusions and Decoctions

Different sea fennel aerial parts were assayed for the providing of infusions and decoctions enriched with antioxidant compounds and chlorogenic acids [23,70,93]. Accordingly, tisanes from sea fennel leaves, flowers, and stems were prepared with infusion and decoction (100 °C) extraction in water for 5 min, resulting in enriched phenolic compounds such as CGA (0.0412 and 0.0434 mg mL^−1^), p-hydroxybenzoic, ferulic acids, epicatechin, pyrocatechol, and 4-hydroxybenzaldehyde, especially from leaves [23]. The authors found that tisane from the leaves and flowers of sea fennel had an excellent scavenging capacity, FRAP, and copper chelating activity and were effective as rooibos tisanes, using a comparison matrix. Similarly, the infusions and decoctions of the aerial parts of sea fennel showed high radical scavenging activity on DPPH (IC_50_: 0.0365 ± 0.0014 and 0.0447 ± 0.0044 mg mL^−1^, respectively) and ABTS (IC_50_: 0.0373 ± 0.0026 and 0.0384 ± 0.0018 mg mL^−1^, respectively) [70]. However, Siracusa et al. [93] found that 81.7% of chlorogenic acid extracted in tisane (0.2 mg mL^−1^) was not stable after two steps of in vitro digestion. Notably, sea fennel was also reported as a diuretic, digestive, and carminative agent [68]. Moreover, melatonin, known to improve rest and sleep [94], was detected for the first time in an extract obtained with an ethanolic solution from sea fennel cultivated in Central Italy [20].

## 6. Exploitation of Sea Fennel for the Production of Nutraceuticals

Sea fennel can also deliver several functional (e.g., anti-inflammatory, antioxidant, and antiatherogenic) properties as a nutraceutical [8,12].

The role of nutraceuticals is to prevent diseases, such as obesity, cardiovascular diseases, and diabetes, and to promote health [95]. Given the unique properties of sea fennel, its exploitation to produce nutraceuticals with neuroprotective, antioxidant, analgesic, immunomodulatory, antimicrobial, antidiabetic, and cardioprotective properties has previously been proposed [11]. The administration of a nutraceutical and dietary supplement should be oriented towards advanced dosage forms such as orodispersible tablets, fast-dissolving films, and easy-swallowing gels for difficult-swallowing subjects [96]. Hence, powders, isolates, and extracts, together with excipients such as gums, chitosan, alginate, and polysaccharides, should be used in the development phase [96].

To exploit sea fennel as nutraceutical, to date, different dosage forms have been evaluated, including isolates (pure and micro-encapsulated) and extracts, as detailed in the following subsections.

### 6.1. Isolates

Isolates are intended for highly concentrated EOs obtained from the aerial parts of sea fennel. Though they are available on the market as commercial products for pharmaceutical purposes, to these authors’ knowledge, isolates have not yet been exploited in food applications.

Steam distillation and hydro-distillation are the most used methods for the extraction of EOs [97]; however, solvent-, microwave-, ultrasound-assisted, and supercritical fluid extraction can also be used [98]. Once isolated, EOs possess great potential in the food sector, mainly due to their antioxidant traits and antimicrobial activity against spoilage and/or pathogenic agents. Scientific evidence and even consumer preference for healthy and natural foods are driving the food industry towards the substitution of synthetic additives, with natural molecules acting as preservatives, such as EOs [99,100,101]. These molecules can be mixed in food formulations [102] or included in controlled atmospheres, films, and coatings [103], meeting the “clean label” trend [99,104]. Petretto et al. [105] applied sea fennel EOs to an experimental device to assess the antimicrobial activity exerted by volatile molecules on food products in a controlled atmosphere. Based on the results of such a study, the vapor phase of sea fennel EOs is enriched in γ-terpinene, α-pinene, p-cymene, sabinene, and α-thujene [105]. Considering the acknowledged antimicrobial [10] and antioxidant activity [106] of γ-terpinene, sea fennel EOs might find an application in the development of controlled atmospheres.

The main drawback of the application of EOs is their solubility, thermal and chemical instability, and volatility. Micro-encapsulation could be an optimal strategy to solve this issue. However, this technique requires delivery systems such as liposomes, cyclodextrins, solid and lipid particles, micelles, and polymer-based carriers to preserve and even enhance the activity of bioactive compounds [42]. Among these delivery systems, micro-emulsion formulation, an aqueous dispersion, is more suitable for scaling-up, and it is easy to handle and formulate, which increases the dispersibility, favoring the precise interaction of EO with the target site. In a recent investigation, polysorbate, glycerol, and ethanol were used to formulate microemulsions for the delivery of sea fennel EOs (particle sizes ranging from 20 to 200 nm), with microencapsulated EOs being more active than pure EOs against fungi (*Candida albicans* and *Microsporum canis*) and bacteria (*E. coli* and *S. aureus*) [41]. Even Campana et al. [42] formulated stable microemulsions of sea fennel EOs with antifungal and antibacterial activities but without using ethyl oleate in the formulation.

### 6.2. Extracts

Sea fennel extracts are to be intended as solid or liquid extracts obtained with solvents, whose production may involve different steps, such as maceration, homogenization, stirring, heating, filtration, stabilization though freeze-drying or drying, and encapsulation, before application in foods [107]. Sea fennel polar extracts, obtained through the separation with hydro-alcoholic solutions, usually target the recovery and concentration of phenolic compounds. The profile of phytochemicals in these extracts depends on the raw material, polarity of the solvent (e.g., ethanol, methanol, water, acetone), the extraction technology (e.g., ultrasounds), and processing parameters (time and temperature conditions), as reported in Table 1. Depending on the solvent polarity, additional compounds besides phenols can also be extracted, including carotenoids (e.g., xanthophylls and carotenes) or vitamin C. As an example, variable levels of carotenoids were found in methanol and chloroform extracts of sea fennel aerial parts [14,63], whereas vitamin C was detected in sea fennel aqueous extracts [22].

To date, the addition of sea fennel extracts to foods as powders [30] or encapsulated forms with liposomes [73] has been explored. In the first study, sunflower oil was added with 20% methanolic and hexane ultrasound-assisted extracts from sea fennel for an enrichment of the bioactive compounds, thus leading to an increase in the oil’s antioxidant capacities (DPPH and FRAP) and content of carotenoids, chlorophylls, phenolics, and flavonoids [30]. In the second investigation, the anti-inflammatory properties, bioaccessibility, and intestinal absorption of sea fennel extract encapsulated in soy phosphatidylcholine liposomes were evaluated, and encouraging results were found [73].

In another study, preparations of freeze-dried soy phosphatidylcholine liposomes encapsulating increasing concentrations of aqueous and ethanolic sea fennel extracts were investigated for their particle properties, water dispersibility, color, thermal properties, and antioxidant capacity (radical scavenging capacity, ferric ion reducing power, Folin-reactive substances) [22]. The same sea fennel liposomes were also investigated for their cytotoxicity to THP-1 and Caco-2 cells and anti-inflammatory effect on THP-1 cells differentiated into macrophages; the chlorogenic acid content after being simulated by in vitro gastrointestinal digestion was also assessed, again with positive results being collected [73].

## 7. Exploitation of Sea Fennel for Production of Edible Films

A further explored application of sea fennel in the food industry deals with the production of edible films destined for the packaging of perishable foods. In fact, Rico et al. [108] evaluated the use of chitosan-based edible film formulated with sea fennel aerial parts and extracts to extend the shelf life of fish burgers. Based on the results collected overall, the incorporation of sea fennel into the films as extracts or treated dried plants did not produce any antimicrobial effect. However, fish burgers coated with a film containing the extract were characterized by a better pH and enhanced antioxidant and lipid oxidation traits. For sensory traits, an opposite trend emerged, with burgers being coated with the film prepared with dried sea fennel showing higher acceptability than those coated with the film containing the extract.

## 8. Conclusions

According to the scientific literature herein reviewed, sea fennel represents a promising vegetable ingredient to produce innovative foods, food ingredients, and nutraceuticals. Nevertheless, further studies are recommended to deepen the knowledge of this emerging crop and, thus, to encourage its large-scale diffusion through an increase in both consumer and farmer awareness about its highly valuable nutritional and functional traits. In this regard, a project titled “Innovative Sustainable Organic Sea Fennel (*Crithmum maritimum* L.)-based Cropping Systems to Boost Agrobiodiversity, Profitability, Circularity, and Resilience to Climate Changes in Mediterranean Small Farms” (acronym SEAFENNEL4MED, https://seafennel4med.com/, accessed on 8 June 2023) has recently been financed within the PRIMA program (call 2021) to introduce new sustainable organic sea-fennel-based cropping systems which are able to cope with limited resources (fresh waters and fertile soils), environmental constrains (biodiversity loss and chemical pollution), and climate-related risks (soil salinization and water drought) for the enhancement of food production stability over time. To this end, laboratory- and industrial-scale prototypes of new high-value foods (e.g., pasta, snacks, and beverages), food ingredients (e.g., condiments), and nutraceuticals not yet available on the market are planned to be formulated and further validated using this highly appreciated halophyte.

## Figures and Tables

**Figure 1 molecules-28-04741-f001:**
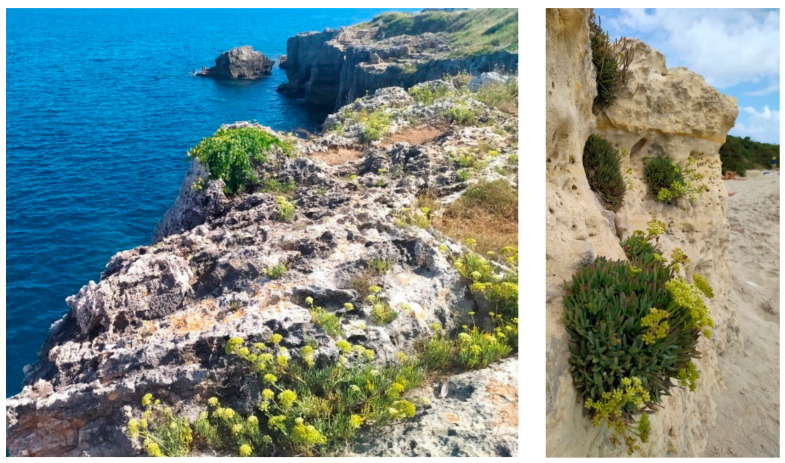
Wild populations of *Crithmum maritimum* L. grown on cliffs (Apulia, Italy).

**Figure 2 molecules-28-04741-f002:**
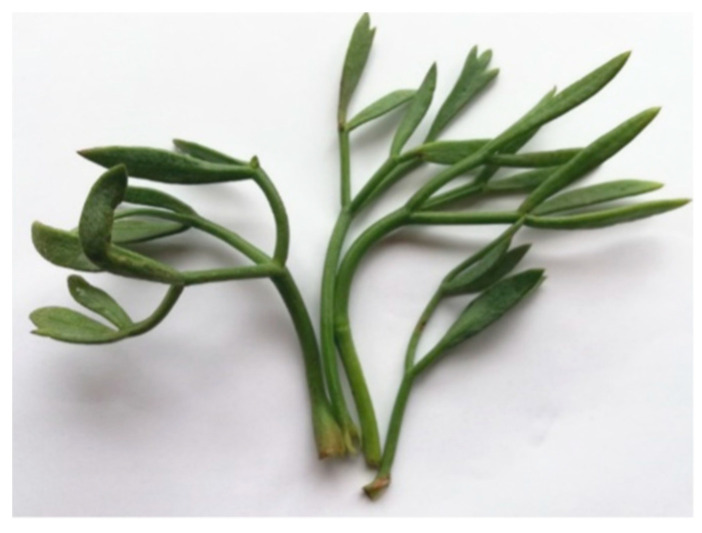
Sea fennel leaves.

**Figure 3 molecules-28-04741-f003:**
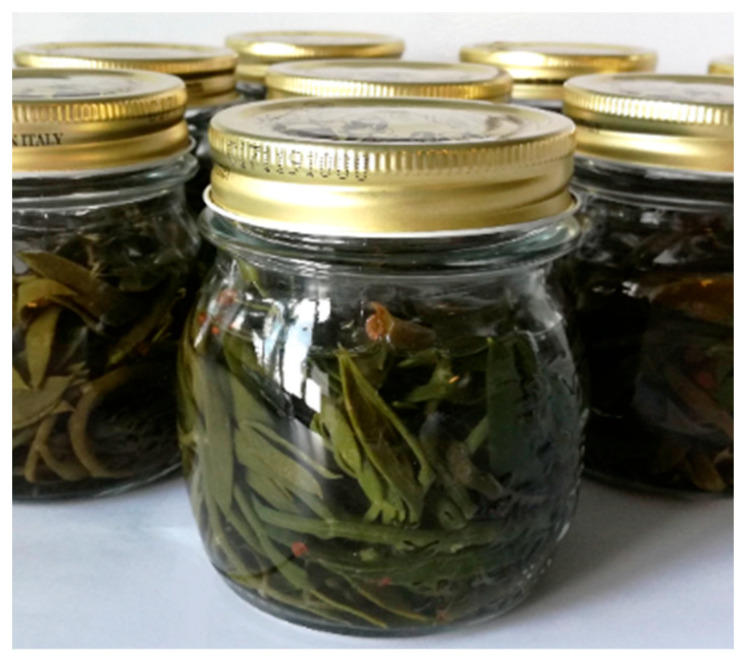
Laboratory-scale prototype of fermented sea fennel preserve.

**Figure 4 molecules-28-04741-f004:**
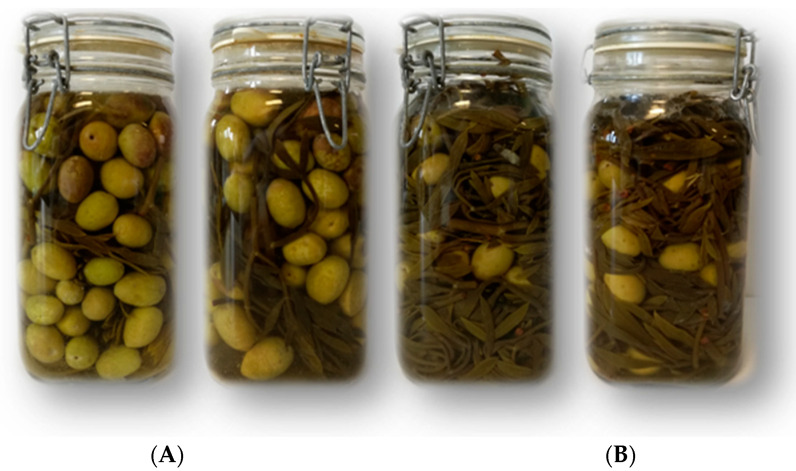
Laboratory-scale prototypes of sea fennel sprouts co-fermented with green olives (*Olea europaea* L. cv. Ascolana tenera) according to two recipes [(**A**): 10% sea fennel and 90% green olives; (**B**): 60% sea fennel and 40% green olives].

**Figure 5 molecules-28-04741-f005:**
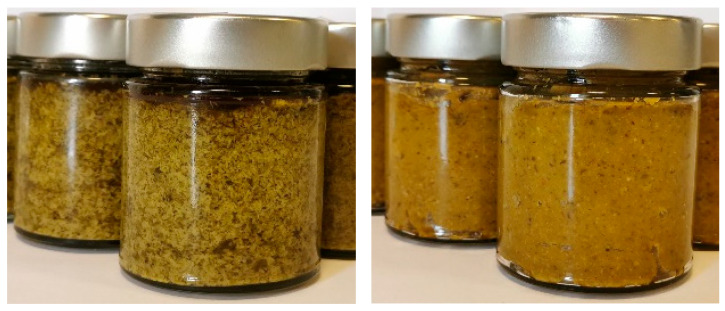
Shelf-stable sea fennel sauces.

**Figure 6 molecules-28-04741-f006:**
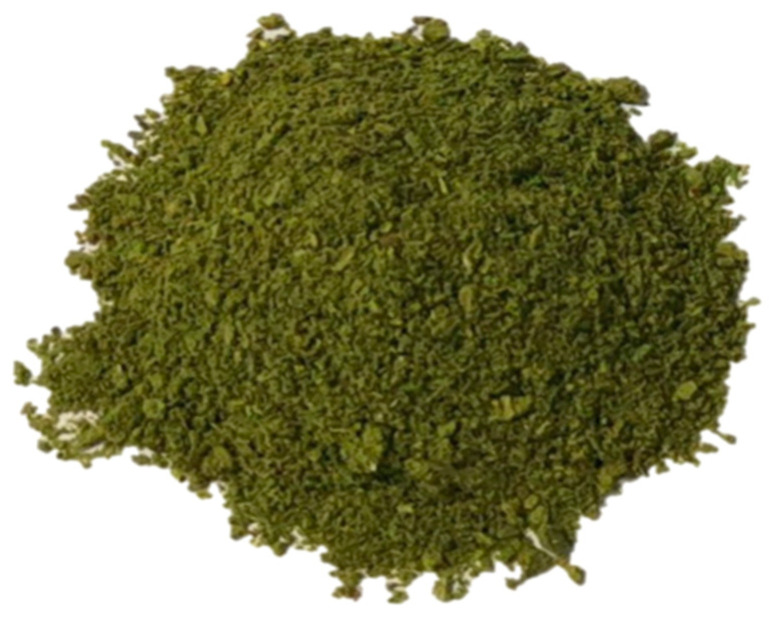
Sea fennel powder.

**Table 3 molecules-28-04741-t003:** Fatty acids main groups composition (% of total fatty acids) in sea fennel.

Analyzed Sample	SFA (%)	MUFA (%)	PUFA (%)	ω6/ω3	Reference
Leaves	23.8–31.2	7.8–8.7	62.5–68.0	1.1–1.4	[51]
	21.0–22.4–21.4	8.3–7.4–8.1	70.6–71.8–70.7	0.9–1.1–1.0	[39]
	11.8–13.8	2.2–2.5	83.1–86.0	1.1–1.2	[14]
	14.1	1.7	84.2	1.3	
	13.3	2.4	84.3	1.4	
	23.9–35.3	15.6–22.0	55.0–60.4	0.8–0.9	[7]
	20.5–24.4	7.1–19.9	33.8–59.4	0.1–0.9	
	28.2–30.1	4.3–25.4	46.4–64.0	0.8–1.1	[9]
	24	18.9	57	0.8	
	24.8–29.2	7.3–7.8	63.6–67.5	1.5	[34]
	22.3–27.9	5.8–9.7	62.3–71.9	1.1–1.6	
	26.0	32.0	42.0	N/A	[2]
Stems	27.6	2.7	72.7	1.1	[6]
Seeds oil	10.8–16.2	72.2–77.8	11.5–11.8	N/A	[32]
	5.5	78.8	15.7	N/A	[50]

SFAs: saturated fatty acids; MUFAs: monounsaturated fatty acids; PUFAs: polyunsaturated fatty acids; N/A: not available.

## Data Availability

Data sharing is not applicable to this article as no new data were created or analyzed in this study.

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
