# Peer review of "Sea Fennel (Crithmum maritimum L.) as an Emerging Crop for the Manufacturing of Innovative Foods and Nutraceuticals"

_molecules, 2023, doi:10.3390/molecules28124741_

Round 1
Reviewer 1 Report
see attached

changes as per attached needed.
Author Response
First we would like to thank the reviewer for his/her valuable suggestions, which helped us to improve our manuscript.
We report below the answers to each Referee's suggestion/remark.
Title: Change to “…as an emerging crop for the manufacturing of…”
ANSWER: correction done
General comments: Manuscript is a review of the chemical composition, extraction schemes and applications of sea fennel as a human food ingredient. It appears to competently cover the current literature of an uncultivated crop of mostly European interest. Some editing of the language is needed.
ANSWER: as suggested the authors will benefit of the editing service offered by MDPI for the revision of the use of English
Line 4: Change “… used for cuisine…” to “used in cuisine…”
ANSWER: correction done
Line 59: Paragraph describes sea fennel as being a protected plant species. Is there any cultivation that would enable wider use in food?
ANSWER: as mentioned by the reviewer and reported in the text, in some geographical area sea fennel is among protected species; however, across the Mediterranean basin, it is currently produced and commercialized by a few crop producers. This information was already reported in the text at lines 58-60 “To date, C. maritimum L. is under-used in Europe and the Mediterranean basin, though small and medium-sized farms increasingly exploit this natural resource for crop production.”
Line 137: Change to read “According to various recent reports [27, 32, 33]…”
ANSWER: correction done
Line 454: Change “At this…” to “In this…”
ANSWER: correction done
Lines 469 to 470: Reword to read “…to adapt by adhering to sea…”
ANSWER: correction done
Line 507: Change “poweders” to “powders”
ANSWER: correction done
Line 515: Change “tratment” to “treatment”
ANSWER: correction done
References 109 and 110 were not cited in the text.
ANSWER: Reference 109 is cited in Table 1, whereas reference 110 is cited in Table 2.
Reviewer 2 Report
Dear Authors,
The manuscript, presented here for consideration and a possible publication in Molecules, is well-structured and organized. Some points must be addressed before it can be recommended for publication. My comments/suggestions are as follows:
1-The abstract should give a full picture of the manuscript and it must contain the following items: Background, objectives, methodology, the main results, conclusions,
2-Please revise the title, "as an emerging" instead of emerging,
3-Before the second "Health beneficial compounds", I suggest another section in which the authors can document the plant biology, its environmental requirements (soil, climate, etc), available genotypes/ecotypes, their potential of valorization, etc.,
4-Discussion must be deepened in light of published literature. Please compare the potentialities of your plants with major Mediterranean Medicinal Aromatic plants. Such comparison can include essential oil main compounds, minerals profiling, phenolics, flavonoids, and antioxidant potential (https://doi.org/10.1155/2023/6308773; https://doi.org/10.1016/j.bcab.2022.102569; https://doi.org/10.1007/s12011-021-03062-w).5-Please keep in mind that a well-designed illustration (table, table, scheme) is worth more than a thousand words. In such a context, along with text, more expressive illustrations can be added. e.g. Paragraph 4.1. can be introduced as a scheme before details in 4.1.1, etc,
6-Please indicate how did you get your bibliographic data (databases, etc),
7-Conclusions section must be rewritten with the main findings and perspectives,
Best regards.
English must be improved.
Author Response
First we would like to thank the reviewer for his/her valuable suggestions, which helped us to improve our manuscript.
We report below our answer to each suggestion/remark.
1-The abstract should give a full picture of the manuscript and it must contain the following items: Background, objectives, methodology, the main results, conclusions.
ANSWER: the abstract has been reorganized and integrated in accordance with the Referee’s suggestion
2-Please revise the title, "as an emerging" instead of emerging,
ANSWER: the title has been modified according to the Reviewer’s suggestion.
3-Before the second "Health beneficial compounds", I suggest another section in which the authors can document the plant biology, its environmental requirements (soil, climate, etc), available genotypes/ecotypes, their potential of valorization, etc.,
ANSWER: All aspects related to sea fennel biology, environmental requirements, available genotypes/ecotypes, and even potential valorization have previously been reviewed by Renna (see for reference:
Renna M. Reviewing the Prospects of Sea Fennel (Crithmum maritimum L.) as Emerging Vegetable Crop. Plants (Basel). 2018 Oct 27;7(4):92. doi: 10.3390/plants7040092. PMID: 30373262; PMCID: PMC6313929)
Hence we intentionally focused our review on the exploitation of sea fennel for production of innovative foods and nutraceuticals, not yet available in the literature.
4-Discussion must be deepened in light of published literature. Please compare the potentialities of your plants with major Mediterranean Medicinal Aromatic plants. Such comparison can include essential oil main compounds, minerals profiling, phenolics, flavonoids, and antioxidant potential (https://doi.org/10.1155/2023/6308773; https://doi.org/10.1016/j.bcab.2022.102569; https://doi.org/10.1007/s12011-021-03062-w).
ANSWER: In the review suggested by the Referee, the sole mineral profiling of twenty wild and cultivated aromatic and medicinal plants growing in Morocco has been discussed; we deliberately focused our review on sea fennel, an emerging crop, by deepening multiple aspects related to both its nutritional and functional traits.
5-Please keep in mind that a well-designed illustration (table, table, scheme) is worth more than a thousand words. In such a context, along with text, more expressive illustrations can be added. e.g. Paragraph 4.1. can be introduced as a scheme before details in 4.1.1, etc,
ANSWER: as even suggested by Referee #3, some of our own pictures (Figure 1 to 5) showing: (i) the growth of wild sea fennel population on cliffs along the seashore; (ii) sea fennel leave; (iii) fermented sea fennel preserve; (iv) co-fermented sea fennel and green olives; (v) sea fennel powder have been added.
6-Please indicate how did you get your bibliographic data (databases, etc).
ANSWER: A section defining databases and search methodology was already reported at the end of the introduction section; web sites of the cited public databases have now been reported for the reader’s convenience.
7-Conclusions section must be rewritten with the main findings and perspectives
ANSWER: as suggested by the Reviewer the conclusion section has been integrated by adding new significant perspectives.
(x) Moderate editing of English language
ANSWER: As suggested by the Reviewer, we will benefit from the MDPI service for editing of the use of English.
Reviewer 3 Report
Introduction
Please add a picture of Crithmum maritimum
Idea in lines 34-36 requires citaton
Please make clearer the idea in lines 55-57
Tables are far from the place from their first citation. Also, tables must be simplified, there are too much repetitive names
Idea in lines 59-60 requires citation
The different subsections (2.2, 2.3, and the others) contain relevant information, however, are not well organized. The authors should present the data more clearly and less redundant. For example: first, a couple of citations on the main polyphenols already found in the plant. Followed by the main methods for their extraction and the methods for their quantification.
Idea in lines 213-215 is not clear. please rewrite it
Section 2.6 must be relocated at the beginning (so it must be section 2.1)
Line 244, please change the word 2 anti-nutrients"
line 245, what is "EO extraction"
why "soluble fiber" is bad?, but in line 257 said is good?
line 252-256 must be rewritten. the idea is not clear enough
Phrase in lines 385-387 is disconnected from the rest of the section
English is fine, there are some minor mistakes that can be easily overcome
Author Response
First we would thank to the reviewer for his/her valuable suggestions, which helped us to improve our manuscript.
We report below the answer to each suggestion/remark.
Reviewer 3
Introduction
Please add a picture of Crithmum maritimum
ANSWER: A picture of Crithmum maritimum L. has been added
Idea in lines 34-36 requires citaton
ANSWER: Citations added
Please make clearer the idea in lines 55-57
ANSWER: the concept has been better clarified.
Tables are far from the place from their first citation. Also, tables must be simplified, there are too much repetitive names.
ANSWER: Tables have been moved close to their first citation. They have also been simplified, by avoiding the repetition about type of analyzed sample.
Idea in lines 59-60 requires citation
ANSWER: the fact that sea fennel crop is actually produced by a few Mediterranean farms has never been reported before, but it is a matter of fact, from a search in the net where various producers emerge (es.: RINCI SrL, https://www.rinci.it/en/.; Kopper cress, https://www.koppertcress.com/en/producten/sea-fennel-r etc).
The different subsections (2.2, 2.3, and the others) contain relevant information, however, are not well organized. The authors should present the data more clearly and less redundant. For example: first, a couple of citations on the main polyphenols already found in the plant. Followed by the main methods for their extraction and the methods for their quantification.
ANSWER: Section 2 was re-organized and improved, especially polyphenol (2.1) and vitamins sections (2.3).
Idea in lines 213-215 is not clear. please rewrite it
ANSWER: the statement has been rephrased.
Section 2.6 must be relocated at the beginning (so it must be section 2.1)
ANSWER: The text of sections 2.6 has been moved at the end of section 2.1.
Line 244, please change the word 2 anti-nutrients"
ANSWER: The word anti-nutrient has been properly used in the text; hence the definition of anti-nutrient is a natural or synthetic compound that interferes with the body's absorption of nutrients.
This term has been used with the same meaning herein specified in multiple published papers (see for instance:
- López-Moreno, M. Garcés-Rimón, M. Miguel,
Antinutrients: Lectins, goitrogens, phytates and oxalates, friends or foe?,
Journal of Functional Foods, Volume 89, 2022, 104938, ISSN 1756-4646, https://doi.org/10.1016/j.jff.2022.104938.
(https://www.sciencedirect.com/science/article/pii/S1756464622000081)
line 245, what is "EO extraction"
ANSWER: EO stands for essential oils; this acronym has been first cited in section 2.4 specifically dedicated to these extracts.
why "soluble fiber" is bad?, but in line 257 said is good?
ANSWER: In contrast to most food, soluble fiber has the rare ability to reduce absorption and digestion. Fiber subtracts rather than adds. In the case of sugars and insulin, this is a beneficial effect; however, dietary fiber also negatively affects the assimilation of minerals, and its excessive dietary intake can lead to gastro-intestinal disorders.
This has been better clarified in the text.
line 252-256 must be rewritten. the idea is not clear enough
ANSWER: the subsection about dietary fibre has been rephrased.
Phrase in lines 385-387 is disconnected from the rest of the section
ANSWER: we disagree with the reviewer since in this statement, the first exploitation of a starter deriving from yogurt has been briefly reported; in the previous statement, fermentation with selected starters is introduced to the reader and described as a powerful tool to produce vegetable preserves from a substrate like sea fennel with antimicrobial properties.
Round 2
Reviewer 2 Report
Dear all,
The comments, raised in the first round, were not taken into account to improve the manuscript.
Regards.
English editing is needed.
Author Response
Dear Reviewer
first we would like to thank you for your effeorts to improve our review; regarding the comment "Extensive editing of English language required", as we anticipated in our answers to the first reviewing round, the manuscript has been throughly revised for the use of English by the MDPI editing service, so we kindly invite you to check the manuscript and verify the significant improvement of the manuscript.
Regarding the comment "The comments, raised in the first round, were not taken into account to improve the manuscript", we confirm that we kindly replied to all the comments raised by reviewer #2 during the first reviewing round. For one of these comments, suggesting to compare the potentialities of sea fennel with major Mediterranean Medicinal Aromatic plants, we kindly replied that this was out of the scope of our review, whose contents were agreed with the editor before drafting the manuscript. We would like also to stress that such a comparison would have significantly lengthened the review, making it worse to read.
For another of the previous comments, suggesting to include a further section to describe the plant biology, its environmental requirements (soil, climate, etc), available genotypes/ecotypes, their potential of valorization, etc.,), we kindly replied that again, this was out of the scope of our review, which was focussed on the application of sea fennel in the food and nutraceutical industries (given the lack of such a review in the available literature); in our reply, we also stressed the fact that a review documenting the plant biology as well as environmental and agronomic requirements is already available in the literature.
We do believe that beyond these two suggestions, which have not taken under consideration (with a clear justificatiuon), we have positively replied to all the remaining issues, which have fully taken under consideration for the improvement of the review.
For your convenience, we report below the original comments and relative answers provided after the first reviewing round (in italic):
1-The abstract should give a full picture of the manuscript and it must contain the following items: Background, objectives, methodology, the main results, conclusions.
ANSWER: the abstract has been reorganized and integrated in accordance with the Referee’s suggestion
2-Please revise the title, "as an emerging" instead of emerging,
ANSWER: the title has been modified according to the Reviewer’s suggestion.
3-Before the second "Health beneficial compounds", I suggest another section in which the authors can document the plant biology, its environmental requirements (soil, climate, etc), available genotypes/ecotypes, their potential of valorization, etc.,
ANSWER: All aspects related to sea fennel biology, environmental requirements, available genotypes/ecotypes, and even potential valorization have previously been reviewed by Renna (see for reference:
Renna M. Reviewing the Prospects of Sea Fennel (Crithmum maritimum L.) as Emerging Vegetable Crop. Plants (Basel). 2018 Oct 27;7(4):92. doi: 10.3390/plants7040092. PMID: 30373262; PMCID: PMC6313929)
Hence we intentionally focused our review on the exploitation of sea fennel for production of innovative foods and nutraceuticals, not yet available in the literature.
4-Discussion must be deepened in light of published literature. Please compare the potentialities of your plants with major Mediterranean Medicinal Aromatic plants. Such comparison can include essential oil main compounds, minerals profiling, phenolics, flavonoids, and antioxidant potential (https://doi.org/10.1155/2023/6308773; https://doi.org/10.1016/j.bcab.2022.102569; https://doi.org/10.1007/s12011-021-03062-w).
ANSWER: In the review suggested by the Referee, the sole mineral profiling of twenty wild and cultivated aromatic and medicinal plants growing in Morocco has been discussed; we deliberately focused our review on sea fennel, an emerging crop, by deepening multiple aspects related to both its nutritional and functional traits.
5-Please keep in mind that a well-designed illustration (table, table, scheme) is worth more than a thousand words. In such a context, along with text, more expressive illustrations can be added. e.g. Paragraph 4.1. can be introduced as a scheme before details in 4.1.1, etc,
ANSWER: as even suggested by Referee #3, some of our own pictures (Figure 1 to 5) showing: (i) the growth of wild sea fennel population on cliffs along the seashore; (ii) sea fennel leave; (iii) fermented sea fennel preserve; (iv) co-fermented sea fennel and green olives; (v) sea fennel powder have been added.
6-Please indicate how did you get your bibliographic data (databases, etc).
ANSWER: A section defining databases and search methodology was already reported at the end of the introduction section; web sites of the cited public databases have now been reported for the reader’s convenience.
7-Conclusions section must be rewritten with the main findings and perspectives
ANSWER: as suggested by the Reviewer the conclusion section has been integrated by adding new significant perspectives.
(x) Moderate editing of English language
ANSWER: As suggested by the Reviewer, we will benefit from the MDPI service for editing of the use of English.
To conclude, we regret the poor general evaluation received in this reviewing round, given the positive comment received in the first round ("The manuscript, presented here for consideration and a possible publication in Molecules, is well-structured and organized").
Kind regards